# When the Majority is Wrong: Modeling Annotator Disagreement for Subjective Tasks

**Eve Fleisig**[†]
UC Berkeley

**Rediet Abebe**
Harvard University

**Dan Klein**
UC Berkeley

## Abstract

Though majority vote among annotators is typically used for ground truth labels in machine learning, annotator disagreement in tasks such as hate speech detection may reflect systematic differences in opinion across groups, not noise. Thus, a crucial problem in hate speech detection is determining if a statement is offensive to the demographic group that it targets, when that group may be a small fraction of the annotator pool. We construct a model that predicts individual annotator ratings on potentially offensive text and combines this information with the predicted target group of the text to predict the ratings of target group members. We show gains across a range of metrics, including raising performance over the baseline by 22% at predicting individual annotators' ratings and by 33% at predicting variance among annotators, which provides a metric for model uncertainty downstream. We find that annotators' ratings can be predicted using their demographic information as well as opinions on online content, and that non-invasive questions on annotators' online experiences minimize the need to collect demographic information when predicting annotators' opinions.

## 1 Introduction

For many machine learning tasks in which the ground truth is clear, having multiple people label examples, then averaging their judgments, is an effective strategy: if nearly all annotators agree on a label, the rest were likely inattentive. However, the assumption that disagreement is noise may no longer hold if the task is more subjective, in the sense that the ground truth label varies by person. For example, the same text may truly be offensive to some people and not to others (Figure 1). Recent work has questioned several assumptions of majority-vote aggregated labels: that there is

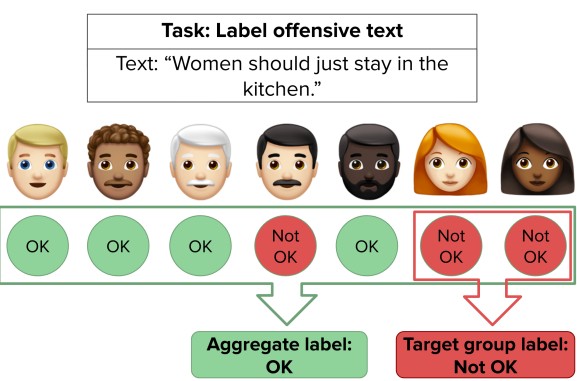

Figure 1: Majority vote aggregation obscures disagreement among annotators due to their lived experiences and other factors. Modeling individual annotator opinions helps to determine when the group targeted by a possibly-hateful statement disagrees with the majority on whether the statement is harmful.

a single, fixed ground truth; that the aggregated judgments of a group of annotators will accurately capture this ground truth; that disagreement among annotators is noise, not signal; that such disagreement is not systematic; and that aggregation is equally suited to very straightforward tasks and more nuanced ones (Larimore et al., 2021; Palomaki et al., 2018; Pavlick and Kwiatkowski, 2019; Prabhakaran et al., 2021; Sap et al., 2022; Waseem, 2016).

In tasks such as hate speech detection, annotator disagreement often results not from noise, but from differences among populations, e.g., demographic groups or political parties (Larimore et al., 2021; Sap et al., 2022). Thus, training with aggregated annotations can cause the opinions of minoritized groups with greater expertise or personal experience to be overlooked (Prabhakaran et al., 2021).

In particular, annotators who are members of the group being targeted by a possibly offensive statement often differ in their opinions from the majority of annotators who rate the text. Because these annotators may have relevant lived experi-

---

[†]Corresponding author: efleisig@berkeley.edu

ence or greater familiarity with the context of the statement, understanding when and why their opinions differ from the majority can help to capture cases where the majority is wrong or opinions are truly divided over whether the example is offensive.

We construct a model that predicts individual annotators' ratings on the offensiveness of text, as well as the potential target group(s)[1] of the text, and combines this information to predict the ratings of target group members on the offensiveness of the example. Compared to a baseline that does not model individual annotators, we raise performance by 22% at predicting individual annotators' ratings, 33% at predicting variance among annotators, and 22% at predicting target group members' aggregate ratings. We find that annotator survey responses and demographic features allow ratings to be modeled effectively, and that survey questions on annotators' online experiences allow annotators' ratings to be predicted while minimizing intrusive collection of demographic information.

## 2 Motivation and Related Work

Our work draws on recent studies that investigate the causes of disagreement among annotators. Pavlick and Kwiatkowski (2019) and Palomaki et al. (2018) note that disagreement among human annotations is not necessarily noise because there is a plausible range of human judgments for some tasks, rather than a single ground truth. Waseem (2016) finds that less experienced annotators are more likely to label items as hate speech than more experienced ones. When evaluating gang-related tweets, domain experts' familiarity with language use in the community resulted in more informed judgments compared to those of annotators without relevant background (Patton et al., 2019). Annotator perception of racism varies with annotator race and political views (Larimore et al., 2021; Sap et al., 2022) and awareness of speaker race and dialect also affects annotation, such that annotators were less likely to find text in African-American English (AAE) offensive if they were told that the text was in AAE or likely written by a Black author (Sap et al., 2019).

In addition, aggregation obscures crucial differences in opinion among annotators. For example, aggregated labels align more with the opinions of

---

[1]Throughout the paper, we use "target group" or "target groups" to mean the demographic groups to which a text may cause harm, i.e., the groups that are the subject of the text's stereotyping, demeaning, or otherwise hateful content.

White annotators than those of Black annotators because Black annotators are underrepresented in typical annotation pools (Prabhakaran et al., 2021).

This motivates our first objective, **identifying crucial examples where target group members disagree**. Disagreement among annotators may stem from a variety of sources, including level of experience, political background, and background experience with the topic on which they are annotating. That is, only some disagreement stems from the annotators who disagree being *better-informed annotators* or *key stakeholders*. We argue that if a statement may stereotype, demean, or otherwise harm a demographic group, members of that demographic group are typically well-suited to determining whether the statement is harmful: both from a normative standpoint, as they would be the group most affected, and because they likely have the most relevant lived experience regarding the harms that the group faces.

However, majority vote aggregation often obscures the opinion of the demographic being targeted by a statement, and so explicitly modeling the opinion of the target group provides a useful source of information for downstream decisions. Thus, finding cases where members of the group targeted by a harmful statement disagree with the majority opinion is crucial to determining whether the majority opinion is actually a useful label on a particular example, and provides a vital source of information for downstream decisions on whether content should be filtered. We address this by constructing a model that predicts the target group(s) of a statement, then predicts the individual ratings of annotators who are target group members (Section 3.1).

Drawing on earlier approaches such as Dawid and Skene (1979), newer work has used disagreement between annotators as a source of information. Fornaciari et al. (2021) use probability distributions over annotator labels as an auxiliary task to reduce model overfitting. Davani et al. (2022) use multi-task models that predict each annotator's rating of an example, then aggregate these separate predictions to produce a final majority-vote decision as well as a measure of uncertainty based on the variance among predicted individual annotators' ratings of the example. Wan et al. (2023) directly predict the degree of disagreement among annotators on an example using annotator demographics, and simulate potential annotators to predict variance in

the degree of disagreement. Gordon et al. (2022) predict individual annotator judgments based on ratings linked to individual annotators and annotators' demographic information. For each example, their system returns a distribution and overall verdict from a "jury" of annotators with demographic characteristics chosen by an evaluator in an interactive system.

These approaches motivate our second objective, **using disagreement as an independent source of information**. Previous work, such as Gordon et al. (2022), uses human-in-the-loop supervision to find alternatives to aggregated majority vote. Since human judgments depend on the person's background and level of experience with the language usage and content of a text, not all evaluators may be able to correctly identify the target group to select an appropriate jury; their own background may also influence their jury selection (e.g., through confirmation bias) and thus the final system judgment. Gordon et al. (2022) investigate the benefits of the interactive approach; we extend on this work to examine the question of how to account for annotator disagreement without human-in-the-loop supervision. Our model directly identifies the target group(s) of a statement and predicts the ratings of target group members, providing an independent source of information about possible errors by the majority of annotators. Our model's predictions can be used as a source of information by human evaluators in conjunction with human-in-the-loop approaches, or function independently (Section 3.2).

Our third objective considers how to **minimize collection of annotators' demographic or identifiable information**. Using individual annotator IDs that map annotators to their ratings permits fine-grained prediction of individual annotator ratings. However, this information is often unavailable (e.g., in the data used by Davani et al., 2022) and its collection may raise privacy concerns for sensitive tasks. Our method uses demographic information and survey responses regarding online preferences to predict annotator ratings without the need to individually link responses to annotators (Section 4.1).

In addition, approaches to predicting annotator opinions that rely on extensive collection of demographic information also raise privacy concerns, particularly when asking about characteristics that are the basis for discrimination in some

environments (e.g., sexual orientation). Thus, an open question is whether annotator ratings may be predicted while using less invasive questions and minimizing unnecessary demographic data collection. We find that survey information about online preferences provides an extremely useful alternate source of information for predicting annotator ratings, while remaining less invasive and thus easier to obtain than extensive demographic questions (Section 4.4).

# 3   Approach

Our approach consists of two connected modules that work in parallel. Given information about the annotator and the text itself, the rating prediction module predicts the rating given by each annotator who labeled a piece of text (Figure 2, in red). For this prediction task, we used a RoBERTa-based module (Liu et al., 2019) initially fine-tuned on the Jigsaw toxicity dataset, which we then fine-tuned on the hate speech detection dataset from Kumar et al. (2021). The target group prediction module predicts the demographic group(s) harmed by the input text (Figure 2, in blue). For this prediction task, we fine-tuned a GPT-2 based module on the text examples and annotated target groups from the Social Bias Frames dataset (Sap et al., 2020). At test time, our model predicts the target group for the input text, then predicts the rating that the members of the target group would give to that text (Figure 2, in purple).

## 3.1   Individual Rating Prediction Module

Motivated by the question of whether information about participants besides directly tracking their judgments (e.g. with IDs) can help to predict participant judgments, we experimented with inclusion of two kinds of annotator information: (1) demographic information and (2) responses to a survey indicating annotator preferences and experiences with online content.

We obtained this information from Kumar et al. (2021)'s dataset, in which each example is labeled by five annotators and each annotator labeled 20 examples. For each annotator label, demographic information is provided on the annotator's race, gender, importance of religion, LGBT status, education, parental status, and political stance. The annotators also completed a brief survey on their preferences regarding online content, including the types of online content they consume (news sites,

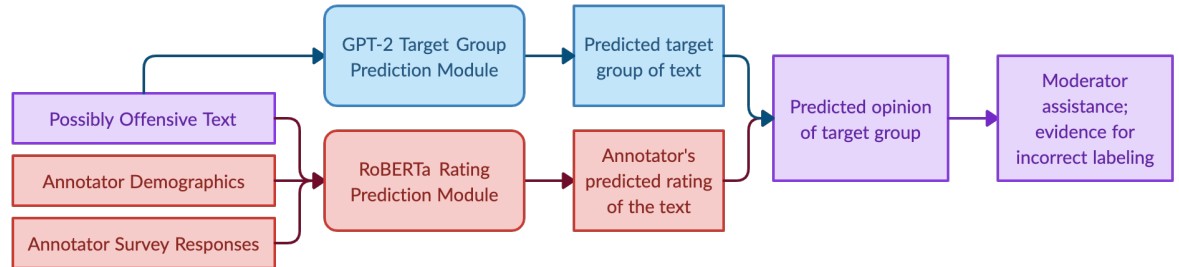

Figure 2: Structure of our approach. Given a piece of text and the annotator's demographic information and survey responses, the rating prediction module predicts the rating given by each annotator who labeled a piece of text (red). The target group prediction module predicts the demographic group(s) harmed by the input text (blue). At test time, our model predicts the target group for the input text, then predicts the rating that the members of the target group would give to that text (purple).

social media, forums, email, and/or messaging); their opinion on how technology impacts people's lives; whether they have seen or been personally targeted by toxic content; and their opinion on whether toxic content is a problem.

For these predictions, the input is formatted as:

$$s_1 \ldots s_n \; [\text{SEP}] \; d_1 \ldots d_n \; [\text{SEP}] \; w_1 \ldots w_n$$

where $s_1 \ldots s_n$ is a template string describing the annotator's survey responses (examples in Appendix A), $d_1 \ldots d_n$ is a template string containing the annotator's demographic information (e.g., "The reader is a 55-64 year old white female who has a bachelor's degree, is politically independent, is a parent, and thinks religion is very important. The reader is straight and cisgender"), $w_1 \ldots w_n$ is the text being rated, and [SEP] is a separator token. We use a template string instead of categorical variables in order to best take advantage of the model's language pretraining objective (e.g., underlying associations about the experiences of different demographic groups).

The model is then trained on the regression task of predicting every annotator's rating (from 0=not at all offensive to 4=very offensive) on each example that they annotated. We use MSE loss with each annotator's rating on an example treated as a separate training point. To compare with the information provided by individual annotator IDs, we additionally trained versions of the model where the input is prepended with an assigned ID corresponding to a unique set of user responses.[2]

At test time, we evaluated performance of this module on the Kumar et al. (2021) dataset using the mean absolute error (MAE) of predicting individual annotators' ratings and MAE of predicting aggregate ratings for an utterance (following Gordon et al., 2022, who evaluated on the same dataset). We also evaluated the MAE of predicting the variance among individual annotator ratings, which provides a measure of uncertainty for model predictions (Davani et al., 2022): if there is high variance among the labels that different annotators are predicted to assign to a piece of text, this can signal that there is low model confidence about whether the text is harmful, or perhaps enough disagreement among the general population that the text should be rerouted to manual content moderation.

For each of these metrics, we hypothesized that modeling individual annotators' ratings with added survey and demographic information would reduce error relative to the baseline, which predicts annotators' aggregate ratings of a piece of text using the average rating as the ground truth (the typical task setup for hate speech detection).

### 3.2 Target Group Prediction Module

The second module predicts the target group(s) of the input text.[3] Following Sap et al. (2020), we used a GPT-2 based module that takes in as input:

$$w_1 \ldots w_n \; [\text{SEP}] \; t_1 \ldots t_n$$

---

[2]The original responses in the dataset do not provide IDs, but we found that fewer than 9% of the unique sets of survey and demographic responses corresponded to more than one annotator, so we assigned IDs to unique sets of responses.

[3]In theory, it is possible to train both modules simultaneously, end-to-end. We trained them separately because at the time of writing, to our knowledge, there was no single dataset that contained sufficiently extensive information on annotators' demographic information, ratings from individual annotators, and labels for the predicted target group of the text.

where $w_1 \ldots w_n$ is the text being rated and $t_1 \ldots t_n$ is a comma-separated list of target groups. During training, this module predicts each token in the sequence conditioned on the previous tokens, using GPT-2's standard language modeling objective. This model was fine-tuned on the text examples and target groups from the Social Bias Frames dataset of hate speech (Sap et al., 2020).

This module generates a free-text string of predicted target groups, used instead of a categorical prediction because there is a long tail of many specific target groups. After the model produces this free-text string, it standardizes the word forms of the list of target groups using a mapping from a list of word forms or variants of a demographic group (e.g., "Hispanic people", "Latinx folks") to one standardized variant ("Hispanic"), taken from the list of standardized demographics in Fleisig et al. (2023). Our model then uses string matching between this list of target groups and the template strings describing annotator demographics to find the set of annotators who are part of any of these target groups.

To provide information about the opinion of the group potentially being harmed by a text example at test time, the model combines the outputs of the target group prediction module and the rating prediction module by (1) predicting the target group of a text example and (2) predicting the individual ratings for all annotators in the predicted target group (Figure 2, in purple). To evaluate model performance, given a text example $x$, we examine the predicted individual ratings for all annotators in the predicted target group of $x$ who provided labels for that example.

Appendices A and B contain additional details on model training for reproducibility.

## 4  Results

The key hypotheses we aimed to test were:

*Can we predict the ratings of individual annotators, using only their demographic information and information about their opinions on online content, without a specific mapping from each annotator to the statements they annotated?* To measure this, we examine how well the individual rating prediction module predicts individual annotators' information given demographic information and/or survey responses, compared to a baseline that does not use information about the annotators (Section 4.1).

*Can we minimize unnecessary collection of in-formation about annotators, while still accurately predicting their ratings?* To examine this, we train models that omit some survey and demographic information and measure their performance to investigate what information contributes most to the accuracy of model predictions (Section 4.4).

Most importantly, given a statement that may target a specific demographic group, *can we predict the ratings of target group members on whether the statement is harmful?* We evaluate the full model's performance at predicting the ratings of target group members on a statement (predicting the target group + predicting the ratings of that group's members). To measure performance on this task, we examine the model error at predicting the ratings given by annotators who labeled examples where they themselves were members of the group targeted by the example (Section 4.3).

### 4.1  Individual Rating Prediction Module

We evaluated the performance of our rating prediction module with different information about the annotators provided as input (combinations of demographic information, survey responses, and annotator IDs). Table 1 gives the mean absolute error on predicting annotators' individual ratings, the aggregate ratings for all annotators who labeled an example, and the variance between annotators labeling an example. We found that both demographic information and survey responses are useful predictors of annotator ratings, as adding either demographic or survey information results in a 10% improvement at predicting individual annotators' ratings. Combining both sources of information results in the most accurate predictions: the model that includes both survey and demographic information raises performance over the baseline by 22% at predicting individual annotators' ratings and 33% at predicting variance among all the annotators who rated an example.

Our best model for individual rating prediction also improves performance at predicting aggregate ratings (by taking the average of the predicted individual annotators' ratings), raising performance by 16% over the baseline. Because this model has access to demographic and survey information about the individual annotators who labeled a response, but the baseline has no information about which subset of annotators labeled an example, this result is unsurprising. However, it further supports the hypothesis that even when predicting the aggregate

| Model | Individual Rating MAE | Aggregate Rating MAE | Variance MAE |
|---|---|---|---|
| Text only | 0.88 | 0.49 | 1.16 |
| + IDs | 0.92 | 0.51 | 1.11 |
| + demographics | 0.79 | 0.44 | 1.01 |
| + surveys | 0.79 | 0.45 | 1.00 |
| + demographics + IDs | 0.85 | 0.46 | 1.02 |
| + demographics + ID + survey | 0.74 | 0.41 | 0.84 |
| **+ demographics + survey** | **0.69** | **0.41** | **0.78** |

Table 1: Mean absolute error of rating prediction module with different input features. Training with demographic and survey information results in the greatest error reduction when predicting individual annotators' ratings; the aggregate ratings of all annotators who labeled an example; and the variance between the annotators who labeled an example. Validation set results can be found in Appendix C.

rating of several annotators on an example, the aggregate rating of that small subgroup of annotators is sufficiently dependent on the subgroup's backgrounds and opinions that taking these into account substantially improves accuracy over predicting the rating of the "average" annotator.

By contrast, inclusion of annotator IDs as a feature may in fact hinder our model's performance. This may be due to the fact that the language model objective does not take full advantage of categorical features and is better suited to textual features. This hypothesis is borne out by previous work on the same dataset: our model achieves lower error than previous work when using only demographic features[4] and the input text (individual annotator MAE = 0.77, compared to 0.81 in Gordon et al., 2022), but not when using demographic features and annotator IDs (individual annotator MAE = 0.80, compared to 0.61 in Gordon et al.). Thus, compared to previous work, this approach appears to make better use of text features, but not of categorical features without meaningful text representations.

## 4.2 Target Group Prediction Module

To evaluate the performance of the target group prediction module on the Kumar et al. (2021) dataset, which does not list the demographic groups targeted by the examples, we manually annotated 100 examples from the dataset's test set with the groups targeted by the text. We then measured the word movers' distance between the predicted text and the freeform text that annotators provided for the

group(s) targeted by the example, the accuracy on exactly matching the list of target groups, and accuracy on partially matching the list of target groups. The model achieves a word movers' distance of 0.370 on the dataset, exact match accuracy of 58%, and partial match accuracy of 81% (error analysis in Appendix C).

## 4.3 Full Model Performance

To evaluate the overall performance of the model (the target group prediction and rating modules together), we had the target group module predict the target groups of the examples in the test set, then had the rating prediction module predict the ratings of members of the target group who rated that example. This is a harder task than predicting individual ratings across the entire dataset, since target group members tend to belong to groups already underrepresented in the annotator pool, such that there is less training data about their behavior.

We considered whether the model can accurately estimate the ratings of target group members, which is necessary to flag cases where the consensus among target group members differs from that of the majority, by defining a *target offense error* metric. Given a set of examples $X$, where each example $x_i$ has a target demographic group[5] $t_i$, we defined the target offense error as the mean absolute error across all $x_i \in X$ of the members of $t_i$ who annotated example $x_i$, where the average rating of the members of $t_i$ who annotated example $x_i$ is at least 1 on a 0-5 scale (i.e., they found the example at least somewhat offensive).

We also evaluated the performance of the com-

---

[4]We exclude queer and trans status from the demographic information here for even comparisons with previous work, which did not include these features.

[5]When there are multiple target groups, we define $t_i$ as the union of those groups.

bined model at predicting the individual and aggregated ratings among annotators who are members of $t_i$, as well as the variance among members of $t_i$ when two or more of them labeled $x_i$ (Table 2). We found that the model achieves an MAE of 0.59 at predicting the average rating of the target group (22% improvement over the baseline), 0.73 at predicting individual annotator ratings for the target group (18% improvement), and target offense error of 0.8 (17% improvement). The model also has an MAE of 1.22 at predicting the variance among target group members (28% improvement). This suggests that the model not only captures differences in opinion between target group members and non-members, but also variation in opinion between members of the target group. This helps to prevent group members from being modeled as a monolith and, by providing a sign of disagreement among experienced stakeholders, provides a particularly useful measure of model uncertainty in hate speech detection.

| Metric (Target Group Only) | Baseline | Best Model (demographics + survey) |
|---|---|---|
| Individual Rating MAE | 0.89 | **0.73** |
| Aggregated Rating MAE | 0.76 | **0.59** |
| Variance MAE | 1.69 | **1.22** |
| Target Offense Error | 0.96 | **0.80** |

Table 2: Combined model performance at predicting opinions of annotators who are members of the target group.

To understand model performance on annotators from different demographics and text targeting different demographics (see also Appendix C), we first examined the five annotator demographic groups with the highest and lowest error rates. We found that the model performs best at predicting the ratings of annotators who are conservative, non-binary, Native American or Alaska Native, Native Hawaiian or Pacific Islanders, or had less than a high school degree (individual rating MAE under .66). The model performed worst at predicting the ratings of annotators who are liberal, politically independent, transgender, had a doctoral or professional degree, or for whom religion was somewhat important (individual rating MAE over 1.24).

In addition, to understand model performance on text targeting different groups, we examined the five target groups with the highest and lowest error rates. We found that individual rating MAE was lowest (under 0.35) at predicting ratings on text targeting people who were racist, Syrian, Brazilian, teenagers, or millennials, and individual rating MAE was highest (over 1.52) at predicting ratings on text targeting people who were well-educated, non-violent, Russian, Australian, or non-believers. To understand effects on text targeting intersectional groups or multiple groups, we examined the performance for different numbers of target groups and found that individual rating MAE changed by no more than 0.02 for text targeting up to four target groups.

Overall, we found that the model is able to combine its prediction of the target group of a statement with predictions of the rating of individual annotators in order to estimate the ratings of target group members on relevant examples, and that it does so best when provided with both demographic information about the annotators and responses to survey questions about their online experiences. We also found that the model accurately predicts target group ratings in the key cases where target group members find that the statement is offensive.

## 4.4 Ablations

The effectiveness of information on annotator demographics and their preferences regarding online content for rating prediction suggests that predicting individual ratings is feasible even when tracking individual annotators' ratings across questions is not possible or excessively obtrusive. However, an additional concern is that asking for some types of demographic information (e.g., sexual orientation, gender identity) or asking for many fine-grained demographic attributes can infringe on annotator privacy, making such data collection both harder to recruit for and ethically fraught. For these reasons, we trained models that omit some of the survey and demographic information to investigate which are most necessary to effectively predict annotator ratings.

By training models on single features, we found that opinions on whether toxic comments are a problem in online settings are the most useful features in isolation for predicting annotator ratings, followed by annotator race (Table 3). Using forward feature selection, we also found that a model trained on only three features–annotators' race, their opinion on whether toxic posts are a problem, and the types of social media they use–

| Feature | Individual Rating MAE | Aggregate Rating MAE | Variance MAE |
|---------|------------------------|------------------------|--------------|
| Gender | 1.0 | 0.67 | 1.2 |
| **Race** | **0.88** | **0.48** | **1.1** |
| LGBT status | 1.0 | 0.67 | 1.2 |
| Education level | 1.0 | 0.57 | 1.2 |
| Age | 0.98 | 0.56 | 1.2 |
| Political leaning | 1.0 | 0.57 | 1.2 |
| Importance of religion | 0.92 | 0.51 | 1.2 |
| Impact of tech | 1.0 | 0.67 | 1.2 |
| Social media use | 1.0 | 0.67 | 1.2 |
| **Thinks toxic posts are a problem?** | **0.87** | **0.49** | **1.1** |
| Personally seen toxic content? | 0.95 | 0.52 | 1.2 |
| **Race + Thinks toxic posts are a problem? + Social media use** | **0.79** | **0.44** | **1.02** |

Table 3: Performance of models trained on single features. Race and opinion on whether online toxic comments are a problem are the most useful individual features for predicting annotator ratings. The model given only annotators' race + opinion on whether toxic posts are a problem + social media usage gives near-identical performance to the model trained with annotators' full demographic information.

approximately matches the performance of the model trained with annotators' full demographic information at predicting annotator ratings (individual rating MAE=0.79). This result suggests that future studies could use more targeted information collection to predict individual annotator ratings, reducing privacy risks while minimizing effects on model quality. Another potential implication is that for studies with limited ability to recruit a perfectly representative population of annotators, it may be most important to prioritize recruiting a representative pool of annotators along the axes that have the greatest effect on annotator ratings, such as race.

Demographic information is useful both to predict individual ratings, and to identify annotators who are relevant stakeholders for a particular information. For the latter purpose, there is an inherent tradeoff between certainty that an annotator is a member of a relevant demographic group, and protecting privacy by not collecting that demographic information. However, the use of non-demographic survey questions may allow practitioners to more

finely tune this tradeoff: survey questions that correlate with demographics (e.g. types of social media used may correlate with age) give partial information without guaranteeing that any single annotator is a member of a specific group. In other words, non-demographic survey questions could function a mechanism to tune the tradeoff between privacy protection and representation. A caveat is that such proxy variables may introduce skews or other issues into data collection, an important issue for future work. Future extensions of this research could also consider whether other specific survey questions are more useful for predicting ratings, allowing for accurate predictions while avoiding unnecessary collection of annotator information.

## 5 Conclusion

We presented a model that predicts individual annotator ratings on the offensiveness of text, which we applied to automatically flagging examples where the opinion of the predicted target group differs from that of the majority. We found that our model raises performance over the baseline by 22% at predicting individual annotators' ratings, 33% at predicting variance among annotators, and 22% at predicting target group members' aggregate ratings. Annotator survey responses and demographic features allow ratings to be modeled without the need for identifying annotator IDs to be tracked, and we found that a model given only annotators' race, opinion on online toxic content, and social media usage performs comparably to one trained with annotators' full demographic information. This suggests that questions about online preferences provide a future avenue for minimizing collection of demographic information when modeling annotator ratings.

Our findings provide a method of modeling individual annotator ratings that can support research on the downstream use of information about annotator disagreement. Predicting the ratings of target group members allows hate speech detection systems to flag examples where the target group disagrees with the majority as potentially mislabeled or as examples that should be routed to human content moderation. Understanding the variance among annotators provides an estimate of model confidence for content moderation or filtering, facilitating a rapid method of spotting cases where a model is uncertain.

In addition, since opinions within a demographic

group are not monolithic, accurately predicting the variance among target group members is especially useful: text that generates disagreement among people who are typically experienced stakeholders may require particularly attentive care. Potential applications of this work that have yet to be explored include improving resource allocation for annotation by identifying the labelers whose judgments it would be most useful to have for a given piece of text, and routing content moderation decisions to experts in particular areas.

## Limitations

This work was conducted on English text that primarily represents varieties of English used in the U.S.; the annotators who labeled the Kumar et al. (2021) dataset were from the United States and the annotators who labeled the Social Bias Frames dataset (Sap et al., 2020) were from the U.S. and Canada. It remains unclear how to generalize this work to groups that face severe harms (e.g., legal discrimination) in the U.S. and Canada or groups that may not face severe harms in the U.S. and Canada, but do face severe harms in other countries. For example, in a country where a group faces serious persecution, it may be unsafe for members of that group from that country to identify themselves and annotate data. In addition, statements that pit perspectives from multiple groups against each other (e.g., text stating that members of one group hurt another group) may be more difficult to analyze under this framework when multiple potential harms towards different groups are at play. Future work could extend this paradigm to uncovering disagreements between multiple groups and refining approaches to intersectional groups.

## Ethical Considerations

Predicting opinions of annotators from different demographic groups runs the risk of imputing an opinion to a group as a monolith instead of understanding the diversity of opinions within that group. Such modeling is no substitute for adequate representation of minoritized groups in data collection. To avoid risks of tokenism, we encourage that individual annotator modeling be used as an aid rather than a replacement for more diverse and participatory data collection. For example, the model can be used to highlight examples for which it would be useful to collect more data from a particular community or delegate decisions to community stake-holders rather than automated decision making. In addition, improving methods for intersectional demographic groups is an important area for future work.

Modeling individual annotators raises new privacy concerns, since sufficiently detailed collection of annotator information may render the annotators re-identifiable. We discuss our findings on some approaches to minimizing privacy concerns in Section 4.4, and as previous work has noted, investigation into techniques such as differential privacy may help to minimize these risks (Gordon et al., 2022).

These questions intertwine with issues of how to collect demographic information, since coarse-grained questionnaires may erase the experiences of more specific or intersectional subgroups, as well as the convergent expertise of different demographic groups that have had overlapping experiences. Further research into effective and inclusive data collection could unlock better ways of understanding these complex experiences.

## Acknowledgments

Many thanks to Deepak Kumar for providing access to the data used in this paper and to Maarten Sap for data and discussions that inspired this research. Thank you to the Berkeley NLP group and recommender systems discussion group for their feedback on this research, and extra thanks to Nicholas Tomlin, Kevin Yang, and Ruiqi Zhong for their especially helpful feedback on earlier drafts.

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

# A  Input Formatting

Example of a sentence representing survey information: "The reader uses social media, news sites, video sites, and web forums. The reader has seen toxic comments, has been personally targeted by toxic comments, thinks technology has a somewhat positive impact on people's lives, and thinks toxic comments are frequently a problem."

Example of a sentence representing demographic information: "The reader is a 55 - 64 year old white female who has a bachelor's degree, is politically independent, is a parent, and thinks religion is very important. The reader is straight and cisgender."

# B  Further Model and Dataset Details

We divided the dataset from Kumar et al. (2021) into a train set of 97,620 examples and validation and test sets of 5,000 examples each (following Gordon et al., 2022, who also use a test set of 5,000 examples for this dataset). We used the Social Bias

Frames dataset's existing split into a training set of 35,424 examples, validation set of 4,666 examples, and test set of 4,691 examples (Sap et al., 2020).

The model uses RoBERTa-base, which has 123 million parameters, and GPT2-large, which has 1.5 billion parameters, for a total of approximately 1.623 billion parameters. No hyperparameter search was conducted; the recommended hyperparameters for RoBERTa-base and GPT2-large were used (Liu et al., 2019; Radford et al., 2018).

For each training run, the model was trained on two NVIDIA Quadro RTX 8000 GPUs for approximately 12 hours.

## C  Results: Details

Table 4 provides the validation results for the metrics in Table 1. Tables 5 and 6 provide details on the demographic groups for which the model performs best and worst.

Overall, each annotator labeled 20 examples on average (stddev=9). For the final level of categories in the ablations in Section 4.4, there are a mean of 31 annotators in each possible bucket (stddev=142).

Examining patterns of error in the target model, we found that the model was most likely to miss a target group when the text was actually targeting people on the basis of race (33% of missed target groups), followed by religion (23%). In cases where the model returned a target group that was not represented in the example, the model most often predicted that the text was targeting people on the basis of race (25% of false positive predictions) or gender (18%).

| Model | Individual Rating MAE | Aggregate Rating MAE | Variance MAE |
|---|---|---|---|
| Text only | 0.87 | 0.48 | 1.16 |
| + IDs | 0.88 | 0.50 | 1.11 |
| + demographics | 0.77 | 0.44 | 1.02 |
| + surveys | 0.77 | 0.43 | 0.98 |
| + demographics + IDs | 0.85 | 0.47 | 1.05 |
| + demographics + ID + survey | 0.72 | 0.41 | 0.89 |
| **+ demographics + survey** | **0.69** | **0.40** | **0.89** |

Table 4: Validation set results for Table 1. Mean absolute error of rating prediction module with different input features.

| Model | Individual Rating MAE |
|---|---|
| Politically conservative | 0.37 |
| Nonbinary | 0.63 |
| American Indian or Alaska Native | 0.64 |
| Native Hawaiian or Pacific Islander | 0.64 |
| Less than a high school degree | 0.66 |
| Religion somewhat important | 0.73 |
| Doctoral degree | 0.73 |
| Transgender | 0.73 |
| Politically independent | 0.83 |
| Politically liberal | 1.24 |

Table 5: Annotator demographic groups with the highest and lowest individual mean absolute error (five best and five worst).

| Model | Individual Rating MAE |
|---|---|
| Racist | 0.06 |
| Syrian | 0.26 |
| Brazilian | 0.28 |
| Teen | 0.33 |
| Millennial | 0.35 |
| Non-believer | 1.52 |
| Australian | 1.52 |
| Russian | 1.52 |
| Non-violent | 1.55 |
| Educated | 1.85 |

Table 6: Predicted target demographic groups with the highest and lowest individual mean absolute error (five best and five worst). Only words that could be mapped to demographic groups are included (non-mappable words such as connectors or stopwords were excluded).