# OpenReview forum: "When the Majority is Wrong: Modeling Annotator Disagreement for Subjective Tasks"
_EMNLP/2023/Conference — EMNLP 2023 Main_

### Official Review · Reviewer_aGsZ · 2023-07-26

**Soundness:** 4

**Excitement:**

4: Strong: This paper deepens the understanding of some phenomenon or lowers the barriers to an existing research direction.

**Missing References:**

None

**Paper Topic And Main Contributions:**

The paper presents a novel approach that jointly 1) infers the target group for a hateful/toxic comment, 2) infers the likely annotation of the message from an annotator in the target group, and 3) uses that information to improve annotation quality according to a number of important metrics

**Questions For The Authors:**

• I’d be curious how coarse-grained that final level of categories is from Section 4.4 that provides good performance with minimal responses too. How many annotators, on average, fall into each unique bucket in the cross product of those categorical variables? A related point - how many responses, on average, did each annotator label, and would we really expect it to be enough for the model to learn something about IDs?

• I don’t understand the result starting at line 414 until the end of the paragraph. If your model is even closer to the aggregate average rating, how is it not even more of a “regression to the majority vote” prediction model? I think there’s an obvious answer here but it’d be helpful to have it spelled out for me, because I can’t quite figure this point out.

• An error analysis of the target group model (Section 4.2) would be useful, even if it is just a few sentences, to get some intuition about where mistakes from the model trend and implications for the overall goal of the paper

• Would be interesting to see Table 2 broken out by major target groups and how that relates to the findings in Table 3, and/or if the model suffers in any cases on particular demographic groups.

**Reasons To Accept:**

+ Thinking about an important problem - how do we perform annotation according to those who are most impacted by the decision?
+ Well-written and intuitive experimental design
+ Some promising results in terms of the selected outcome metrics
+ I do like the *idea* of being about to anonymize annotators to just their responses on some very coarse, non-invasive survey questions, although I have some qualms with how this is framed in the paper (see below).

**Reasons To Reject:**

-  I am a little surprised at how limited the limitations statement in this paper is. First, I think there’s a significant risk of tokenism here. How do we avoid the situations where we just say “well, we had some female annotators, so we are ‘all good on gender bias’”? Or even worse, “we have a model of what a woman would say about this message”? Second, although less important (because I think the work implicitly could address this concern upon extension), the limitations section seems to emphasize cultural differences but only touches, implicitly, on intersectionality. Worth a more explicit nod about how this paper does or does not address it.
- I'm a bit dubious about the idea that this method increases privacy, could make things easier for researchers, or could be used to reduce information about annotators, which seems to be one of the main claims of the article. Specifically, the authors state that a limit to prior work is the need for annotator IDs that “might not be available”, but then suggest adding new survey data that almost certainly is unavailable in prior work.  It’s also not entirely clear to me that the claim that their work can a priori limit invasive data collection is valid.  If I want to ensure that I am giving adequate voice to target populations, it would seem that I have to have some way I can validate that I am doing so, or have a strong proxy variable for it (which negates the claims of privacy preservation).  Even more simply, annotator IDs are in any reasonably ethical research entirely anonymized. It’s not clear to me then how limiting use of annotator IDs creates more private data.


**Reproducibility:**

4: Could mostly reproduce the results, but there may be some variation because of sample variance or minor variations in their interpretation of the protocol or method.

**Reviewer Confidence:**

4: Quite sure. I tried to check the important points carefully. It's unlikely, though conceivable, that I missed something that should affect my ratings.

---

> ### Author Rebuttal · Authors · 2023-08-29
>
> Thank you very much for providing helpful feedback on our paper.
>
> > "I am a little surprised at how limited the limitations statement in this paper is. First, I think there’s a significant risk of tokenism here. How do we avoid the situations where we just say “well, we had some female annotators, so we are ‘all good on gender bias’”? Or even worse, “we have a model of what a woman would say about this message”?
> Second, although less important (because I think the work implicitly could address this concern upon extension), the limitations section seems to emphasize cultural differences but only touches, implicitly, on intersectionality. Worth a more explicit nod about how this paper does or does not address it."
>
> These are important points; thank you very much for raising them. Regarding the risk of tokenism: We agree that our model is no substitute for adequate representation of minoritized groups in data collection. We will highlight the risk of using our model to impute an opinion to a group as a monolith instead of making the effort to examine the diversity of opinions within that group. We will note this as a limitation, and emphasize that the model should be used as an aid rather than a replacement for more diverse and participatory data collection. For example, the model could be used to highlight examples for which it would be useful to collect more data from a particular community or delegate decisions to community stakeholders rather than automated decision making.
>
> Regarding intersectionality: we will (a) highlight this as a limitation of the paper and (b) expand on the discussion of cases in which the model works and does not work, with particular emphasis on intersectional groups, in the camera-ready version. We will discuss the performance at predicting ratings on different targeted demographic groups, and at predicting ratings by members of different demographic groups, outlined briefly here:
>
> We found that the model performs best at predicting the ratings of annotators who are conservative, nonbinary, Native American or Alaska Native, Native Hawaiian or Pacific Islanders, or had less than a high school degree (individual rating MAE ranging from .37 to .66). The model performed worst at predicting the ratings of annotators who are liberal, politically independent, transgender, had a doctoral or professional degree, or for whom religion was somewhat important (individual rating MAE ranging from .73 to 1.24).
>
> We also found that individual rating MAE was lowest (.06 to 0.35) at predicting ratings on text targeting people who were racist, Syrian, Brazilian, or teenagers, and individual rating MAE was highest (1.2 to 1.85) at predicting ratings on text targeting people who were well-educated, non-violent, Russian, or Australian. To understand effects on text targeting intersectional groups or multiple groups, we examined the performance for different numbers of target groups and found that individual rating MAE dropped by no more than 0.02 for text targeting up to four target groups.
>
> We will discuss these findings in the paper (with a table of the full demographic breakdowns in the appendix).
>
>
> > “I'm a bit dubious about the idea that this method increases privacy, could make things easier for researchers, or could be used to reduce information about annotators, which seems to be one of the main claims of the article. Specifically, the authors state that a limit to prior work is the need for annotator IDs that “might not be available”, but then suggest adding new survey data that almost certainly is unavailable in prior work…Even more simply, annotator IDs are in any reasonably ethical research entirely anonymized. It’s not clear to me then how limiting use of annotator IDs creates more private data.”
>
> We will clarify this further for the camera-ready, but we would like to separate out the issues of (a) creating a system that does not require annotator IDs and (b) the potential uses of including survey data.
>
> Not requiring annotator IDs is helpful because existing datasets in which annotator information has been removed for anonymity (e.g., the data used by Davani et al., 2022 or the Annotators with Attitudes dataset from Sap et al., 2022) often have the following format:
>
> - [example A] [label 1; annotator demographics] [label 2; annotator demographics] [label 3; annotator demographics]
> - [example B] [label 1; annotator demographics] [label 2; annotator demographics] [label 3; annotator demographics]
>
> e.g.:
> - [“I hate group X”] [5; 35 year old White man] [4; 55 year old Black man] [5; 25 year old Hispanic woman]
> - [“I hate group Y”] [1; 45 year old Hispanic man] [3; 35 year old White man] [5; 60 year old White woman]
>
> This format leaves it unclear whether the 35-year-old white men who labeled “I hate group X” and “I hate group Y” are the same person. Unlike prior work, our approach allows annotators to be modeled without this information. We apologize for the lack of clarity on the distinction between questions of privacy, and questions of addressing current data limitations. We will clarify that modeling without annotator IDs is more related to limitations of how data collection is presently done, rather than privacy.
>
> Meanwhile, we emphasize the potential usefulness of survey information as an alternative to collecting demographic information (separate from the question of annotator IDs). Collecting demographic data improves modeling of annotators, but such data collection can be highly invasive. If we can model annotator opinions just as well by knowing a proxy variable from survey data, this could help to preserve annotator privacy.
>
> > “It’s also not entirely clear to me that the claim that their work can a priori limit invasive data collection is valid. If I want to ensure that I am giving adequate voice to target populations, it would seem that I have to have some way I can validate that I am doing so, or have a strong proxy variable for it (which negates the claims of privacy preservation).”
>
> Yes, this is an excellent point and we will discuss this issue in the camera-ready version. There is an intrinsic tradeoff (not limited to our work, but in annotator modeling overall) between certainty that the data is representative of a group G, and privacy protection that limits knowledge of who is in G. At minimum, we would like to note that such a tradeoff exists, and that use of non-demographic survey questions allows practitioners to choose points along this spectrum besides “full demographic information” and “no demographic information.” For example, if labeling a transphobic comment ideally requires transgender annotators, but collecting such information is highly sensitive, then asking whether an annotator has faced gender-based discrimination online will result in a subset of annotators with a higher percentage of transgender annotators but no guarantee that any single annotator is transgender. In other words, non-demographic survey questions can be a mechanism to “tune” the tradeoff between privacy protection and representation. (Investigating whether the use of different proxy variables introduces other issues into data collection is an important question for future work.)
>
> > “I’d be curious how coarse-grained that final level of categories is from Section 4.4 that provides good performance with minimal responses too. How many annotators, on average, fall into each unique bucket in the cross product of those categorical variables? A related point - how many responses, on average, did each annotator label, and would we really expect it to be enough for the model to learn something about IDs?”
>
> For the final level of categories in Section 4.4, there are a mean of 31 annotators in each possible bucket (stddev=142). Overall, each annotator labeled 20 examples on average (stddev=9).
>
> > I don’t understand the result starting at line 414 until the end of the paragraph. If your model is even closer to the aggregate average rating, how is it not even more of a “regression to the majority vote” prediction model? I think there’s an obvious answer here but it’d be helpful to have it spelled out for me, because I can’t quite figure this point out.
>
> Suppose a dataset has six annotators A,B,C,D,E,F, and example 1 is labeled by annotators A,B,C. The baseline model doesn’t know which subset of annotators labeled example 1, so its prediction for the aggregate rating of A,B,C would be the same as its prediction for the aggregate rating of A,B,C,D,E,F. However, our best model predicts the individual ratings of A,B,C and then averages them when predicting the aggregate rating. If the average of A,B,C is far from the average of A,B,C,D,E,F, then our model would outperform the baseline model at predicting the aggregate rating on example 1. We will clarify this in the paper.
>
> > “An error analysis of the target group model (Section 4.2) would be useful, even if it is just a few sentences, to get some intuition about where mistakes from the model trend and implications for the overall goal of the paper. Would be interesting to see Table 2 broken out by major target groups and how that relates to the findings in Table 3, and/or if the model suffers in any cases on particular demographic groups.
>
> Thank you; we will expand on the error analysis of the target group model as suggested for the camera-ready version.

---

### Official Review · Reviewer_7dom · 2023-07-31

**Soundness:** 4

**Excitement:**

4: Strong: This paper deepens the understanding of some phenomenon or lowers the barriers to an existing research direction.

**Missing References:**

-

**Paper Topic And Main Contributions:**

The authors construct a model that predicts individual annotator ratings on possibly problematic language and then combine this knowledge with the text's projected target group to predict target group members' ratings. When comparing MAE (Mean Absolute Error) to a baseline that does not model individual annotators, the authors find that the new model outperforms the baseline by 22% in predicting individual annotators' ratings, 33% in predicting variance among annotators, and 22% in predicting aggregate ratings of target group members.

**Questions For The Authors:**

How can those who organize annotation sessions introduce this study in their future annotation sessions?

**Reasons To Accept:**

The paper is very well written, and it addresses a crucial and common sense topic in human data annotation, which leads to the fact that, at the end of the day, different annotators' backgrounds influence the quality of our machine learning models.

**Reasons To Reject:**

-

**Reproducibility:**

4: Could mostly reproduce the results, but there may be some variation because of sample variance or minor variations in their interpretation of the protocol or method.

**Reviewer Confidence:**

2: Willing to defend my evaluation, but it is fairly likely that I missed some details, didn't understand some central points, or can't be sure about the novelty of the work.

**Typos Grammar Style And Presentation Improvements:**

-

---

> ### Author Rebuttal · Authors · 2023-08-29
>
> Thank you very much for providing feedback on our paper. Regarding your question, we will release our code with the final version of the paper so that the model can be more widely used by future researchers conducting annotation work.

---

### Official Review · Reviewer_Muv1 · 2023-08-05

**Soundness:** 4

**Excitement:**

4: Strong: This paper deepens the understanding of some phenomenon or lowers the barriers to an existing research direction.

**Paper Topic And Main Contributions:**

The authors present an interesting dimension for being inclusive of the majority as well as the minority opinion. It is a critical line of work in the dimension of ethics and ML. The paper paper presents a differentiated model training for identifying the targeted demographics in hate speech and offensive language.

**Questions For The Authors:**

A. The dataset by Sap et al., Social Bias Inference Corpus is a good test subject for the work presented since it does not have a lot examples but the content is severely subjective and toxic, any reason as to why it is not considered?
B. How does your work go against a non demographic modeling method such as the Dawid & Skyene approach?


**Reasons To Accept:**

A. The model architecture the authors have presented is quite interesting. The decision for train/fine-tune each head on a dataset that is relevant to it is an interesting approach.
B. The paper is easy to read and follow, well motivated.

**Reasons To Reject:**

A. The results in the tables do support the model does work, but paper seems to lack empirical results as to where each case on what works and what doesn't. The authors have an interesting example at the beginning (Fig 1) to demonstrate their motivation, but this wasn't highlighted well enough in their later findings. Not even the appendix. I feel this was a missed opportunity for a submission to a venue like EMNLP.

See the questions.

**Reproducibility:**

5: Could easily reproduce the results.

**Reviewer Confidence:**

4: Quite sure. I tried to check the important points carefully. It's unlikely, though conceivable, that I missed something that should affect my ratings.

**Typos Grammar Style And Presentation Improvements:**

A. The limitations and the ethics sections does not count towards the 8 page limit, feels like it could have been better used for more empirical results.

---

> ### Author Rebuttal · Authors · 2023-08-29
>
> Thank you very much for providing helpful feedback on our paper.
>
> > “The results in the tables do support the model does work, but paper seems to lack empirical results as to where each case on what works and what doesn't.”
>
> We will expand on the discussion of cases in which the model works and does not work in the camera-ready version, with details on the performance at predicting ratings on different targeted demographic groups, and at predicting ratings by members of different demographic groups:
>
> We found that the model performs best at predicting the ratings of annotators who are conservative, nonbinary, Native American or Alaska Native, Native Hawaiian or Pacific Islanders, or had less than a high school degree (individual rating MAE ranging from .37 to .66). The model performed worst at predicting the ratings of annotators who are liberal, politically independent, transgender, had a doctoral or professional degree, or for whom religion was somewhat important (individual rating MAE ranging from .73 to 1.24).
>
> We also found that individual rating MAE was lowest (.06 to 0.35) at predicting ratings on text targeting people who were racist, Syrian, Brazilian, or teenagers, and individual rating MAE was highest (1.2 to 1.85) at predicting ratings on text targeting people who were well-educated, non-violent, Russian, or Australian. To understand effects on text targeting intersectional groups or multiple groups, we examined the performance for different numbers of target groups and found that individual rating MAE dropped by no more than 0.02 for text targeting up to four target groups.
>
> We will discuss these findings in the paper (with a table of the full demographic breakdowns in the appendix).
>
> > “The dataset by Sap et al., Social Bias Inference Corpus is a good test subject for the work presented since it does not have a lot examples but the content is severely subjective and toxic, any reason as to why it is not considered? B. How does your work go against a non demographic modeling method such as the Dawid & Skyene approach?”
>
> We used the SBIC data as a test set for the target group model, but not the full model, because compared to the Kumar et al. dataset, the SBIC dataset has (1) significantly less demographic information on its annotators, (2) fewer annotators per example, and (3) a much more uneven distribution of annotators per example (many annotators only labeled a single example).
>
> We were unaware of the Dawid-Skene model, but will cite it and discuss differences in the approaches for the camera-ready version.

---

### Meta-Review · Area_Chair_KK5z · 2023-09-13

**Recommendation:** 5

**Metareview:**

This paper studies how modeling annotator individuality creates more accurate classifier as well as how to remedy the subsequent privacy concerns. on All reviewers reach consensus that this paper is sound and exciting. The authors engaged in an in-depth discussion with reviewer aGsZ on this paper's implications and made convincing arguments about its claims. The paper is well-written and easy to follow, and potentially an impactful contribution to the community.

---

### Decision · Program_Chairs · 2023-10-07

**Decision:**

Accept-Main

**Comment:**

This paper studies how modeling annotator individuality creates more accurate classifier as well as how to remedy the subsequent privacy concerns. on All reviewers reach consensus that this paper is sound and exciting. The authors engaged in an in-depth discussion with reviewer aGsZ on this paper's implications and made convincing arguments about its claims. The paper is well-written and easy to follow, and potentially an impactful contribution to the community.